

# Evaluation of the reliability and quality of YouTube videos as a source of information for transcutaneous electrical nerve stimulation

Yüksel Erkin[1], Volkan Hanci[2] and Erkan Ozduran[3]

[1] Anesthesiology and Reanimation, Algology, Dokuz Eylül University, Izmir, Turkey
[2] Anesthesiology and Reanimation, Subdivision of Critical Care Medicine and Resuscitation, Dokuz Eylül University, Izmir, Turkey
[3] Physical Medicine and Rehabilitation, Algology, Dokuz Eylül University, Izmir, Turkey

Corresponding author
Erkan Ozduran,
erkanozduran@gmail.com

## ABSTRACT

**Background**. YouTube plays an influential role in disseminating health-related information in the digital age. This study aimed to evaluate YouTube videos on transcutaneous electrical nerve stimulation (TENS) in terms of their information value and quality.

**Methods**. In this descriptive study, we ranked the first 100 videos that met the inclusion criteria using the search term "transcutaneous electrical nerve stimulation" on October 30, 2022. These videos were classified according to the number of views, likes, dislikes, comments, duration, popularity and content categories. Reliability, quality, and accuracy of the videos were assessed using the Journal of American Medical Association (JAMA) Benchmark Criteria and Modified DISCERN Questionnaire and Global Quality Score (GQS). Video popularity were calculated by the Video Power Index (VPI).

**Results**. Based on the GQS results, we found that 59, 27, and 14 videos had low, intermediate, and high quality, respectively. In addition, based on the JAMA results, 79 and 21 videos had poor and high reliability, respectively. No statistically significant difference was found between the JAMA, modified DISCERN and GQS scores in terms of the sources of videos ($p = 0.226$, $p = 0.115$, $p = 0.812$). Notably, there was a weak positive correlation between the JAMA scores and the number of views ($r = 0.204$, $p = 0.041$).

**Conclusion**. According to our study results, most YouTube videos on TENS were of low quality and reliability. Additionally, most videos were uploaded from sources created by doctors; the most frequently found content was about the TENS procedure, and content on complications of the procedure became less frequent as the videos became more recent. In particular, it was found that videos uploaded by academics have longer duration. It has been found that reliable videos with high JAMA scores also have high number of views. Accordingly, it can be concluded that videos with higher quality and more reliability that are created by healthcare providers will be more useful for patients seeking information about TENS.

## INTRODUCTION

Pain is known to be a major global health problem, and it is estimated that one in five adults suffers from pain annually and one in 10 adults are diagnosed with chronic pain (*Goldberg & McGee, 2011*). In addition, pain is associated with financial burden in the form of medical consultations, treatment, and time lost at work, and it represents a high social burden in the form of suffering and loss of quality of life (*Gaskin & Richard, 2012*). Acute pain is defined as pain of recent onset and probably limited duration, which usually has an identifiable temporal and causal relationship with the injury or disease, and lasts for <3 months (*Johnson et al., 2015*). Chronic pain lasting >3 months is estimated to occur in approximately 50% of the adult population, of which approximately 10%–20% people experience clinically significant pain (*Gibson et al., 2019*). In Europe, 19% of adults report suffering from moderate-to-severe long-lasting pain that negatively affects their social and work life; moreover, they report receiving inadequate management for the same (*Reid et al., 2011*). The presence of chronic pain leads to a poor quality of life as well as reduced productivity, participation, and use of healthcare services, which further have economic implications (*Gibson et al., 2019*).

Although patients with pain are often managed with pharmacological treatment, lack of efficacy and presence of side effects prevent the continuation of treatment. In such patients, nonpharmacological interventions, such as physical therapy and exercise, and electrical stimulation techniques are beneficial (*Wu et al., 2018*). Notably, the transcutaneous electrical nerve stimulation (TENS) device has a portable and battery-powered system that induces electrical stimulation *via* electrodes placed on the skin surface (*Johnson et al., 2015*). Notably, self-administration of TENS is a treatment method that is often preferred by patients as the TENS device can be sold over the counter; patients do not need to go to clinics for treatment; and the procedure produces minimal side effects, toxicity, and overdose (*Chipchase, Williams & Robertson, 2009*).

TENS is primarily used to relieve various types of pain (*e.g.*, neuropathic, nociceptive, and nociplastic). TENS exerts its effects by reducing pain perception through excitatory and inhibitory synapses in the dorsal horn of the spinal cord *via* the pain-gate mechanism, as proposed by Mellzack and Wall in the mid-1960s (*Gibson, Wand & O'Connell, 2017*). Neuropathic pain is seen in approximately 6.9–10% of the general population and significantly affects activities of daily living and quality of life (*Van Hecke et al., 2014*). Many studies have been published on the efficacy of TENS in neuropathic pain. Although the overall quality of the studies is low, there are studies reporting the superiority of TENS in neuropathic pain compared to sham practice (*Gibson, Wand & O'Connell, 2017*). It has been reported in the literature that TENS can be effective in neuropathic pain such as spinal cord injury, lumbar radicular pain, diabetes and postherpetic neuralgia (*Mokhtari et al., 2020*). It is stated that future multicenter clinical trials using optimized TENS protocols may provide more information on the clinical use of neuropathic pain (*Gibson, Wand & O'Connell, 2017*).

Nociplastic pain, seen in 2% of the general population and is another area of use for TENS. This type of pain is seen in clinical conditions such as fibromyalgia, burning mouth

and vulvodynia (*Moisset, Lanteri-Minet & Fontaine, 2020*). In the review by *Arienti (2019)*, TENS has been reported to be effective at low level of evidence in fibromyalgia patients. It was stated that better pain relief could be achieved especially in the group that received exercise and TENS compared to the group that received only exercise.

TENS also has areas of use other than pain symptoms. It is used in the treatment of incontinence by neuroregulation of supraspinal reflex and spinal reflex centers and in the treatment of peripheral ischemia, wound healing, tissue regeneration with its vasodilating effect (*Guo et al., 2014*). In addition, it is stated that TENS improves cognitive symptoms of dementia, as it helps increase brain activity and regeneration with released neurotransmitters (*Cameron, Lonergan & Lee, 2003*). It reduces spasticity in neurological diseases such as stroke by modulation of reciprocal inhibition, reduction of stretch reflex excitability and increase of presynaptic inhibition (*Mahmood et al., 2019*)

It is generally known that patients often search for medical information online in order to better understand their diagnosis, select treatments, and play a more active part in their own healthcare (*Crutchfield et al., 2021*). Notably, Google is the most frequently visited website, followed by the popular video platform YouTube (*Chan et al., 2021*). YouTube, the most popular online platform for sharing non-peer-reviewed videos among US adults, was regularly visited by 73% of US adults and 91% of those aged 15–29 years in 2019–2020 (*Crutchfield et al., 2021*). In addition, approximately eight of 10 Internet users obtain their health information online, and YouTube is one of the most popular websites, with approximately one billion hours of video watched by one billion users per day (*Lee et al., 2020*). Although YouTube is very popular among users, there are concerns about the quality and reliability of the content of videos uploaded on YouTube. Moreover, because videos can be uploaded by anyone without verification and because numerous videos are made for commercial purposes, the content, the information provided, and the accuracy of this information are questionable (*Chang & Park, 2021*). A review of 18 studies examining YouTube videos on a variety of topics found that YouTube provides high-quality health-related information, but it also provides conflicting and misleading health-related information (*Madathil et al., 2015*).

The fact that the TENS device can be bought by users without a prescription, that it can be easily ordered on the Internet, and that the instructions for use must be read and understood, poses some difficulties in its use. In addition, the reliability and quality of the answers in the videos to the questions about TENS, which are consulted by both patients and health professionals, are questionable. There are doubts about whether patients have access to accurate information and how useful videos are for the professional development of health professionals. The primary aim of the present study was to evaluate the quality and reliability of YouTube videos on TENS. The secondary aim was to identify the sources that upload high-quality and more reliable videos. In addition, we compared user parameters between videos of high and low quality or reliability.

## MATERIALS & METHODS

### Ethical approval

For this descriptive content analysis study, ethics approval was obtained from the Dokuz Eylul University Ethics Committee (Decision number: 7548-GOA 2022/34-23, Date: 26.10.2022).

### Analysis of content

On October 30, 2022, two independent reviewers (E.O. and V.H.) identified videos containing medical content using the search term "transcutaneous electrical nerve stimulation" on the YouTube (https://www.youtube.com/) search engine. Further, a neutral term was used to create a broader pool of videos. In the event of disagreements between the two authors in the video evaluation, a third independent author (Y.E.) made the final decision. Notably, the Google Incognito window was used for the study to avoid bias due to search history and cookies. Videos with TENS-related content of acceptable audiovisual quality (without audio–video problems preventing accurate assessment) and those published in English were included in the present study. In contrast, non-English videos, videos without auditory or visual stimuli, and repetitive videos were excluded from the study (*Lee et al., 2020*). These videos were listed using the sort by view count option. Furthermore, similar to other studies in the relevant literature, the first 100 eligible videos were included in the present study after a sensitive evaluation of inclusion and exclusion criteria (*Basch et al., 2021*; *Manchaiah et al., 2020*).

The subject of content in the videos was evaluated based on the presence or absence of the following five TENS-related factors in each video: indications, contraindications, procedures, complications, and types of TENS.

### Reliability assesment

The modified DISCERN scale was developed to assess the reliability of written health information. The assessment of this scale is based on five yes/no questions. The total score is obtained by summing the scores of the yes answers (yes = 1, no = 0), and it ranges from 0 to 5. Higher scores represent higher reliability (*Chang & Park, 2021*) (Table 1).

The Journal of American Medical Association (JAMA) Benchmark is another reliability scale that examines online videos and resources based on four criteria: authorship, disclosure, currency, and attribution. It is used to assess the accuracy and reliability of the relevant videos. Each video was evaluated based on the four abovementioned criteria, and 1 point was awarded for each criterion found in the video. The final score ranges from 0 to 4. Higher scores represent higher reliability (*Silberg, Lundberg & Musacchio, 1997*) (Table 1). According to the JAMA results, videos with a score of 4, 2–3, and 0–1 have completely sufficient, partially sufficient, and insufficient data, respectively (*Ozduran & Büyükçoban, 2022*). In addition, websites with the JAMA scores of $\geq 3$ were considered to have high reliability, and those with the scores of $\leq 2$ were considered to have low reliability (*Silberg, Lundberg & Musacchio, 1997*).

**Table 1  Contents of GQS, DISCERN and JAMA assessment criteria.**

| JAMA Benchmark Criteria | Total Score (0–4 Points) |
|---|---|
| Authorship | 1 point (Authors and contributors, their affiliations,and relevant credentials should be provided) |
| Attribution | 1 point (References and sources for all content should be listed) |
| Disclosure | 1 point (Conflicts of interest, funding,sponsorship, advertising, support, and video ownership should be fully disclosed) |
| Currency | 1 point (Dates that on which the content was posted and updated should be indicated). JAMA is used to evaluate the accuracy and reliability of information) |
| | |
| Modified DISCERN Criteria | Total Score (0–5 Points) |
| 1 Does the video address areas of contro-versy/uncertainty? | 0–1 point |
| 2 Are additional sources of information listed for patient reference? | 0–1 point |
| 3 Is the provided information balanced and unbiased? | 0–1 point |
| 4 Are valid sources cited? | 0–1 point |
| 5 Is the video clear, concise, and understandable? | 0–1 point |
| GQS | Score |
| Poor quality, poor flow of the site, most information missing, not at all useful for patients | 1 |
| Generally poor quality and poor flow, some information listed but many important topics missing, of very limited use to patients | 2 |
| Moderate quality, suboptimal flow, some important information is adequately discussed but others poorly discussed, somewhat useful for patients | 3 |
| Good quality and generally good flow, most of the relevant information is listed, but some topics not covered, useful for patients | 4 |
| Excellent quality and excellent flow, very useful for patients | 5 |

**Notes.**
JAMA, Journal of American Medical Association; GQS, Global Quality Score.

### Quality assesment

Global Quality Score (GQS) is a 5-point Likert scale used to assess the quality of videos. GQS scores range from 1 to 5, and higher scores represent higher quality. Videos with a score of 1 or 2, 3, and 4 or 5 are considered to have low, intermediate, and high quality, respectively (*Chang & Park, 2021*) (Table 1). In case of a discrepancy between the two authors' score, a final score was assigned by a third independent author (Y.E.).

### Assesment of user engagement (Video parameters)

For each video, the following five different engagement metrics were recorded: views, likes, dislikes, video duration (in seconds), and comments. The impact and popularity of the videos were determined using the Video Power Index (VPI). (VPI = like count/(like count + dislike count) × 100) (*Reina-Varona et al., 2022*).

### Sources of videos

Video sources were categorized as academic sources, society/professional organizations, physicians, commercial and health-related websites. In addition, the country and continent

of the upload were recorded. Further, the videos were evaluated based on whether they contained animation.

## Statistical analysis

Acquired data were analyzed using the Statistical Package for Social Sciences (SPSs Inc., Chicago, IL, USA) 24.0 software. Data indicating frequency were expressed as number ($n$) and percentage (%), and continuous variables were expressed as mean ± standard deviation. Kolmogorov–Smirnov and Shapiro–Wilk tests were used to assess whether the data showed a normal distribution pattern. Mann–Whitney U and Kruskal–Wallis tests were used to analyze continuous variables between the study groups, whereas Pearson's chi-square test was used to analyze categorical variables between the study groups. Spearman's correlation test was used to analyze the correlation between variables. A $p$-value of <0.05 was considered to indicate a significant difference.

## RESULTS

After entering the search term, the first 100 videos were included in the study after excluding 14 irrelevant videos (not addressing the topic), 21 non-English videos, 18 repetitive videos, and 17 videos with no sound or video content from the total 170 videos (Fig. 1). If multiple copies of a video were found to have been uploaded, one video was included in the rating, and the others were excluded. A total of 13 h, 43 min, and 56 s of footage was watched. Notably, the length of the shortest video was 23 s, whereas that of the longest video was 1 h and 7 min. The video with the fewest likes received 0 likes, whereas that with the highest likes received 7700 likes. The video with the fewest dislikes received 0 dislikes, whereas that with the highest dislikes received 675 dislikes. The most-viewed video was viewed 2588822 times, and the least-viewed video was viewed six times. The video with the fewest comments received 0 comments, whereas that with the highest comments received 685 comments. The average number of views, comments, likes, dislikes and VPI score per video was 89790.40 ± 290630.04, 45.16 ± 95.43, 503.25 ± 1003.51, 34.73 ± 85.47 and 94.96 ± 6.33, respectively, and the average video duration was 494.36 ± 522.31 s. Notably, 12 (12%) videos had animations. Additionally, 56 (56%) videos were uploaded in 2019 or later. No significant change in animation content presence was observed over the years ($p = 0.469$). Although there was no statistically significant difference in the number of videos with animations by continent ($p = 0.303$), a significant difference was found in the number of videos with animations by country ($p < 0.001$). Accordingly, 41.7% of the videos with animated content were from the US, whereas no animated content was found in videos from India (Table 2).

Regarding the evaluation of videos according to their sources, it was found that 40 (40%) were uploaded by physicians, 25 (25%) by health-related websites, 11 (11%) society/professional organizations, 13 (13%) by commercial websites and 10 (10%) videos by academic sources (Fig. 2).

Of the included videos, 79 (79%), 31 (31%), 30 (30%), 81 (81%), and 44 (44%) contained information on indication, contraindications, complications, procedures, and TENS types, respectively. In particular, when analyzing the content categories by year, videos with

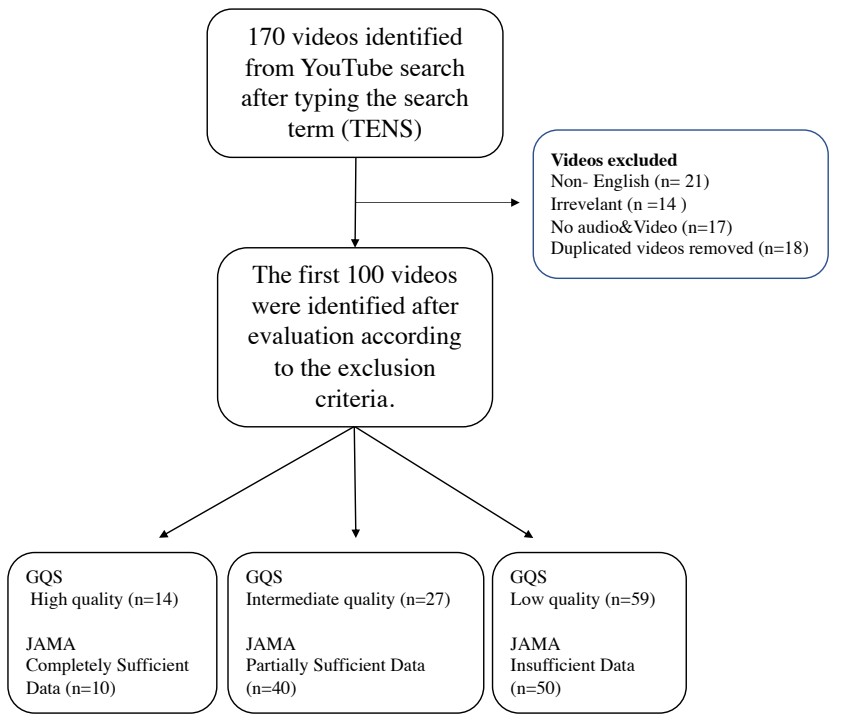

**Figure 1** Flow diagram for review of YouTube videos on TENS.

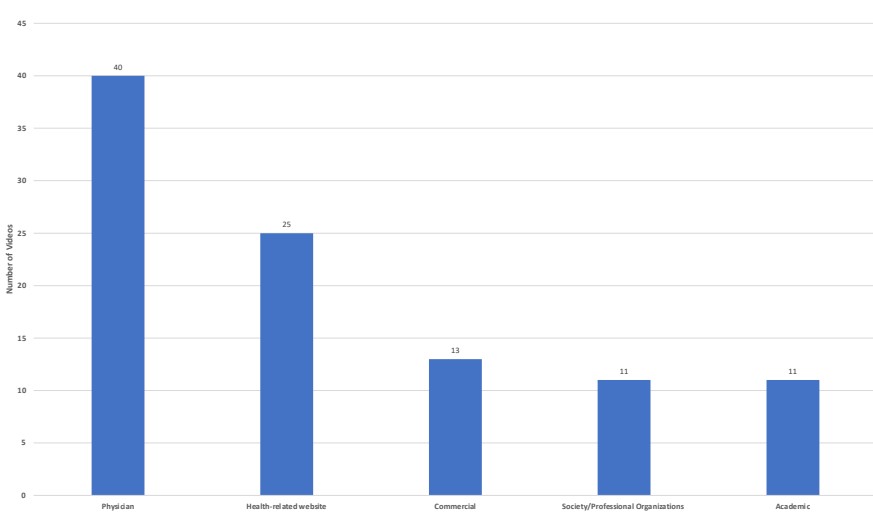

**Figure 2** Video sources.

information on complications were uploaded more frequently in 2018 or previous years ($p = 0.026$). No significant difference between years was found between the other content categories (Table 2).

**Table 2  Comparison of the content, reliability and quality of videos over the years.**

| Video content/ years | | 2022–2019, n (%) | 2018–2015, n (%) | 2014–2011, n (%) | p |
|---|---|---|---|---|---|
| Animation | + | 8 (66.7%) | 4 (33.3%) | 0 (0%) | 0.469 |
| | – | 48 (54.5%) | 31 (35.2%) | 9 (10.2%) | |
| Indications | + | 48 (60.8%) | 23 (29.1%) | 8 (10.1%) | 0.056 |
| | – | 8 (38.1%) | 12 (57.1%) | 1 (4.8%) | |
| Contraindications | + | 20 (64.5%) | 9 (29%) | 2 (6.5%) | 0.506 |
| | – | 36 (52.2%) | 26 (37.7%) | 7 (10.1%) | |
| Complications | + | 11 (36.7%) | 14 (46.7%) | 5 (16.7%) | **0.026** |
| | – | 45 (64.3%) | 21 (30%) | 4 (5.7%) | |
| Tens Types | + | 29 (65.9%) | 11 (25%) | 4 (9.1%) | 0.163 |
| | – | 27 (48.2%) | 24 (42.9%) | 5 (8.9%) | |
| Procedure | + | 47 (58%) | 29 (35.8%) | 5 (6.2%) | 0.124 |
| | – | 9 (47.4%) | 6 (31.6%) | 4 (21.1%) | |
| JAMA | Insufficient data (1 Point) | 27 (54%) | 19 (38%) | 4 (8%) | 0.977 |
| | Partially sufficient data (2 or 3 points) | 23 (57.5%) | 13 (32.5%) | 4 (10%) | |
| | Completely sufficient data (4 points) | 6 (60%) | 3 (30%) | 1 (10%) | |
| JAMA | Low Reliable | 46 (58.2%) | 28 (35.4%) | 5 (6.3) | 0.189 |
| | High Reliable | 10 (47.6%) | 7 (33.3%) | 4 (19%) | |
| GQS | Low quality (1 or 2 points) | 32 (54.2%) | 22 (37.3%) | 5 (8.5%) | 0.777 |
| | Intermediate quality (3 points) | 14 (51.9%) | 10 (37%) | 3 (11.1%) | |
| | High quality (4–5 points) | 10 (71.4%) | 3 (21.4%) | 1 (7.1%) | |
| Modified DISCERN | Very Poor | 14 (60.9%) | 8 (34.8%) | 1 (4.3%) | 0.299 |
| | Poor | 25 (54.3%) | 17 (37%) | 4 (8.7%) | |
| | Fair | 8 (57.1%) | 6 (42.9%) | 0 (0%) | |
| | Good | 7 (46.7%) | 4 (26.7%) | 4 (26.7%) | |
| | Excellent | 2 (100%) | 0 (0%) | 0 (0%) | |

**Notes.**
Pearson Chi-square test.
JAMA, Journal of American Medical Association benchmark criteria; GQS, Global Quality Score.
Bold font: statistical significance.

According to the results of GQS used for quality assessment, 59, 27, and 14 videos were determined as low, intermediate quality, and high quality, respectively. We found that the mean GQS score was $2.42 \pm 0.95$. In addition, based on the GQS, JAMA and DISCERN results, no significant difference was found between the video sources ($p = 0.812$, $p = 0.226$, $p = 0.115$).

Overall, 35.3% of videos with good and excellent reliability were uploaded by physicians, whereas 30.4% of videos with very poor and poor reliability were uploaded by health-related websites (Table 3). When video parameters were evaluated according to the video source, no statistically significant difference was found except duration (view, $p = 0.256$; like, $p = 0.108$; comment, $p = 0.198$; dislike, $p = 0474$; VPI, $p = 0.446$; duration, $p < 0.001$). Accordingly, videos with a long duration are mostly of academic origin. In addition, no statistically significant difference was found when video parameters were evaluated
according to video quality scores (GQS) (view, $p = 0.855$; like, $p = 0.948$; comment, $p = 0.355$; duration, $p = 0.963$; dislike, $p = 0.996$; VPI, $p = 0.919$). When video parameters were evaluated according to the grouping based on JAMA scores (insufficient, partially sufficient, and completely sufficient), a statistically significant difference was found in the number of likes and comments (view, $p = 0.062$; like, $p = 0.035$; comment, $p = 0.045$; duration, $p = 0.306$; dislike, $p = 0.058$; VPI, $p = 0.264$). Accordingly, videos with high reliability received more likes and comments (Table 4).

We found that the mean JAMA score was $1.63 \pm 1.19$. According to the JAMA scores, 10 (10%), 40 (40%), and 50 (50%) videos had completely sufficient, partially sufficient, and insufficient data. According to the JAMA results, 79 (79%) videos had low reliability, and the remaining 21 (21%) videos had high reliability. According to the DISCERN results, 23, 46, 14, 15, and 2 videos were of very poor, poor, fair, good, and excellent quality, respectively. The mean DISCERN score was $2.27 \pm 1.04$.

A weak positive correlation was found between the JAMA scores and views ($r = 0.204$, $p = 0.041$). A moderate positive correlation was found between the JAMA and modified DISCERN scores ($r = 0.649$, $p < 0.001$). A moderate to strong correlation was found between VPI scores and the number of views ($r = -0.589$, $p < 0.001$), likes ($r = -0.485$, $p < 0.001$), dislikes ($r = -0.767$, $p < 0.001$) and comments ($r = -0.459$, $p < 0.001$). No correlation was found between VPI scores and rating scales (JAMA, $r = 0.037$, $p = 0.717$; DISCERN, $r = 0.017$, $p = 0.870$; GQS, $r = -0.045$, $p = 0.658$) (Table 5).

A statistically significant difference was found between the continents where the videos were uploaded in terms of the number of views ($p = 0.001$), likes ($p < 0.001$), dislikes ($p = 0.001$), comments ($p = 0.001$), and JAMA scores ($p = 0.002$), but there was no difference between the continents in terms of duration ($p = 0.129$). When the videos in the study were sorted by country, they were determined as follows; USA 68, India 11, United Kingdom 10 and other countries 11 (Canada, three; Australia, two; Germany, two; Philippines, one; Ireland, one; Japan, one; Pakistan, one). A statistically significant difference was found between the countries where the videos were uploaded in terms of the number of views ($p = 0.008$), likes ($p = 0.005$), dislikes ($p = 0.016$), comments ($p = 0.033$), VPI scores ($p = 0.020$) and JAMA scores ($p = 0.024$) as well as in terms of the animation content ($p < 0.001$) (Table 6). Videos from the USA performed better than videos from other countries in all of the above parameters.

## DISCUSSION

In the present study, YouTube videos on TENS, which patients can use for many indications, especially pain symptoms, were evaluated according to content categories, quality, reliability, and user interaction criteria. For the study, we focused only on TENS, which is one of the noninvasive methods in pain management. A 2016 systematic review of 13 studies involving 267 patients reported that TENS led to significant improvements in pain lasting 2–24 weeks (*Jauregui et al., 2016*). TENS is preferred by patients because it is available without a prescription, is inexpensive, and has a good safety profile compared with drugs (*Johnson et al., 2019*). In the present study, we evaluated the quality and reliability

Erkin et al. (2023), *PeerJ*, DOI 10.7717/peerj.15412

**Table 3  Video sources by reliability and quality parameters.**

| | | Academic (*n*) | Physician (*n*) | Society/ Professional Organization (*n*) | Health-related Website (*n*) | Commercial (*n*) | *p* |
|---|---|---|---|---|---|---|---|
| GQS (1–5 points) | Low quality (1or 2 points) | 7 (11.9%) | 23 (39%) | 7 (11.9%) | 16 (27.1%) | 6 (10.2%) | |
| | Intermediate quality (3 points) | 4 (14.8%) | 12 (44.4%) | 2 (7.4%) | 5 (18.5%) | 4 (14.8%) | 0.812 |
| | High quality (4–5 points) | 0 (0%) | 5 (35.7%) | 2 (14.3%) | 4 (28.6%) | 3 (21.4%) | |
| JAMA score (0–4 Points) | Insufficient data (1 Point) | 4 (8%) | 20 (40%) | 5 (10%) | 13 (26%) | 8 (16%) | |
| | Partially sufficient data (2 or 3 points) | 6 (15%) | 14 (35%) | 3 (7.5%) | 12 (30%) | 5 (12%) | 0.226 |
| | Completely sufficient data (4 points) | 1 (10%) | 6 (60%) | 3 (30%) | 0 (0%) | 0 (0%) | |
| Modified DISCERN score (0–5 points) | Very Poor | 1 (4.3%) | 9 (39.1%) | 1 (4.3%) | 7 (30.4%) | 5 (21.7%) | |
| | Poor | 6 (13%) | 15 (32.6%) | 6 (13%) | 14 (30.4%) | 5 (10.9%) | |
| | Fair | 0 (0%) | 10 (71.4%) | 0 (0%) | 2 (14.3%) | 2 (14.3%) | 0.115 |
| | Good | 3 (20%) | 6 (40%) | 3 (20%) | 2 (13.3%) | 1 (6.7%) | |
| | Excellent | 1 (50%) | 0 (0%) | 1 (50%) | 0 (0%) | 0 (0%) | |

**Notes.**

Pearson Chi-square test.

GQS, Global Quality Score; JAMA, Journal of American Medical Association benchmark criteria.

Erkin et al. (2023), *PeerJ*, DOI 10.7717/peerj.15412

**Table 4  Video parameters according to years, reliability and quality parameters (mean ± standard deviation).**

| Years | View Mean ± SD | Like Mean ± SD | Dislike Mean ± SD | Comment Mean ± SD | Duration Mean ± SD | VPI Mean ± SD |
|---|---|---|---|---|---|---|
| 2019–2022 (n = 56) | 32422.12 ± 72983.73 | 366.35 ± 646.59 | 18.78 ± 45.57 | 30.16 ± 50.09 | 570.68 ± 631.31 | 97.25 ± 4.39 |
| 2015–2018 (n = 35) | 189614.08 ± 466371.56 | 804.91 ± 1440.27 | 60.71 ± 124.24 | 72.57 ± 140.64 | 393.86 ± 258.52 | 91.73 ± 7.12 |
| 2011–2014 (n = 9) | 58545.33 ± 128556.08 | 181.89 ± 355.14 | 32.89 ± 76.06 | 31.89 ± 79.54 | 410.33 ± 508.81 | 93.26 ± 7.83 |
| *p* | **0.002** | 0.113 | **0.001** | 0.266 | 0.224 | **0.001>** |
| Video Sources | | | | | | |
| Academic (n = 11) | 24450.09 ± 49562.94 | 189.27 ± 281.82 | 11.91 ± 24 | 17.81 ± 29.45 | 883.73 ± 1110.45 | 96.61 ± 5.76 |
| Physician (n = 40) | 141087.63 ± 420355.54 | 686.95 ± 899.16 | 41.10 ± 68.29 | 61.65 ± 92.47 | 558.90 ± 412.045 | 96.09 ± 3.85 |
| Society/Professional Organization (n = 11) | 57603.09 ± 116498.01 | 214.64 ± 350.86 | 27.64 ± 69.04 | 30 ± 72.74 | 286.73 ± 170.75 | 95.87 ± 5.84 |
| Health-related Website (n = 25) | 78847.36 ± 212925.39 | 643.24 ± 1586.95 | 48.60 ± 139.47 | 56.76 ± 138.47 | 483.20 ± 396.622 | 93.08 ± 8.26 |
| Commercial (13) | 35520.46 ± 38287.02 | 178.69 ± 201.79 | 13.77 ± 14.87 | 8.07 ± 17.16 | 163.47 ± 145.26 | 92.95 ± 8.37 |
| *p* | 0.256 | 0.108 | 0.474 | 0.198 | **0.001>** | 0.446 |
| GQS (1–5 points) | | | | | | |
| Low quality (1 or 2 points) (n = 59) | 114420.93 ± 368950.72 | 536.36 ± 1130.95 | 39.49 ± 102.17 | 48.52 ± 103.02 | 453.78 ± 371.66 | 94.74 ± 7.12 |
| Intermediate quality (3 points) (n = 27) | 45687.89 ± 80595.92 | 442.07 ± 808.13 | 24.74 ± 49.91 | 33.77 ± 60.57 | 621.11 ± 820.08 | 95.18 ± 4.75 |
| High quality (4–5 points) (n = 14) | 71045.14 ± 125505.41 | 481.71 ± 801.87 | 33.93 ± 61.37 | 52.92 ± 119.49 | 420.93 ± 263.66 | 95.44 ± 5.72 |
| *P* | 0.855 | 0.948 | 0.996 | 0.355 | 0.963 | 0.919 |

Erkin et al. (2023), *PeerJ*, DOI 10.7717/peerj.15412

**Table 4** (*continued*)

| Years | View<br>Mean ± SD | Like<br>Mean ± SD | Dislike<br>Mean ± SD | Comment<br>Mean ± SD | Duration<br>Mean ± SD | VPI<br>Mean ± SD |
|---|---|---|---|---|---|---|
| JAMA score (0–4 Points) | | | | | | |
| Insufficient data (1 Point) ($n = 50$) | 28771.58 ± 52048.25 | 234.52 ± 390.28 | 14.74 ± 30.88 | 19.26 ± 32.76 | 447.54 ± 443.62 | 94.61 ± 6.75 |
| Partially sufficient data (2 or 3 points) ($n = 40$) | 146813.52 ± 435319.85 | 722.25 ± 1372.50 | 45.65 ± 113.05 | 65.30 ± 133.05 | 561.78 ± 655.82 | 95.75 ± 6.04 |
| Completely sufficient data (4 points) ($n = 10$) | 166792 ± 226897.57 | 970.90 ± 1108.05 | 91 ± 116.28 | 94.10 ± 97.08 | 458.80 ± 178.451 | 93.52 ± 5.37 |
| *p* | 0.062 | **0.035** | 0.058 | **0.045** | 0.306 | 0.264 |
| Modified DISCERN score (0–5 points) | | | | | | |
| Very Poor ($n = 23$) | 28291.69 ± 37514.82 | 262.86 ± 378.82 | 13.52 ± 29.41 | 21.86 ± 36.76 | 452.96 ± 488.86 | 96.00 ± 4.60 |
| Poor ($n = 46$) | 55288.28 ± 101387.47 | 399.69 ± 692.69 | 26.28 ± 47.91 | 36.52 ± 78.44 | 480.26 ± 400.484 | 94.01 ± 7.65 |
| Fair ($n = 14$) | 363039.79 ± 706618.22 | 1636.92 ± 1999.83 | 112.86 ± 185.76 | 138.14 ± 174.02 | 504.50 ± 307.92 | 95.11 ± 3.20 |
| Good ($n = 15$) | 46513.20 ± 102512.49 | 195.40 ± 299.45 | 24.87 ± 59.41 | 26.60 ± 62.74 | 602.87 ± 964.58 | 95.47 ± 6.60 |
| Excellent ($n = 2$) | 2407.50 ± 3349.56 | 22.50 ± 30.40 | 0.0 ± 0.0 | 0.0 ± 0.0 | 410 ± 182.43 | 100.00 ± 0.0 |
| *p* | **0.042** | **0.007** | **0.015** | **0.004** | 0.766 | 0.318 |

**Notes.**

Kruskal Wallis Test.

N, Number of videos; SD, Standart Deviation; GQS, Global Quality Score; JAMA, Journal of American Medical Association benchmark criteria; VPI, Video Power Index.

Bold font: statistical significance ($p < 0.05$).

Erkin et al. (2023), *PeerJ*, DOI 10.7717/peerj.15412

**Table 5 Correlation analysis.**

| | GQS | | JAMA | | Modified DISCERN | | Number of views | | Number of likes | | Number of dislikes | | Number of comments | | Video duration | |
|---|---|---|---|---|---|---|---|---|---|---|---|---|---|---|---|---|
| | r | p | r | p | r | p | r | p | r | p | r | p | r | p | r | p |
| JAMA | −0.059 | 0.558 | | | | | | | | | | | | | | |
| Modified DISCERN | −0.018 | 0.859 | 0.649* | **0.001>** | | | | | | | | | | | | |
| Number of views | −0.044 | 0.663 | 0.204* | **0.041** | 0.057 | 0.576 | | | | | | | | | | |
| Number of likes | −0.079 | 0.436 | 0.190 | 0.259 | 0.048 | 0.638 | 0.886 | **0.001>** | | | | | | | | |
| Number of dislikes | 0.005 | 0.958 | 0.140 | 0.165 | 0.083 | 0.412 | 0.877 | **0.001>** | 0.865 | **0.001>** | | | | | | |
| Number of comments | 0.160 | 0.112 | 0.134 | 0.184 | 0.088 | 0.384 | 0.789 | **0.001>** | 0.875 | **0.001>** | 0.794 | **0.001>** | | | | |
| Video duration;second | −0.054 | 0.596 | 0.109 | 0.278 | 0.093 | 0.358 | 0.134 | 0.183 | 0.329 | **0.001>** | 0.230 | **0.021** | 0.330 | **0.001** | | |
| VPI | −0.045; | 0.658 | 0.037 | 0.717 | 0.017 | 0.870 | −0.589; | **0.001>** | −0.485 | **0.001>** | −0.767 | **0.001>** | −0.459 | **0.001>** | −0.017 | 0.864 |

**Notes.**

Spearman's correlation test.

GQS, Global Quality Score; JAMA, Journal of American Medical Association benchmark criteria; VPI, Video Power Index.

Bold font: statistically significant.

Erkin et al. (2023), *PeerJ*, DOI 10.7717/peerj.15412

Peer*J*

**Table 6  Video parameters by country and continent.**

| | View Mean ± SD | Like Mean ± SD | Dislike Mean ± SD | Comment Mean ± SD | Time Mean ± SD | VPI Mean ± SD | Animation n (%) | JAMA Mean ± SD | DISCERN Mean ± SD | GQS Mean ± SD |
|---|---|---|---|---|---|---|---|---|---|---|
| **Continent** | | | | | | | | | | |
| America (n = 71) | 120746.44 ± 340283.37 | 678.13 ± 1167.57 | 46.81 ± 98.86 | 59.34 ± 110.33 | 510.06 ± 534.70 | 95.17 ± 4.82 | 7 (58.3%) | 1.83 ± 1.15 | 36.79 ± 15.58 | 2.47 ± 0.98 |
| Non-America (n = 29) | 14001.48 ± 28036.97 | 148.63 ± 280.69 | 5.13 ± 10.88 | 15.03 ± 37.40 | 461 ± 501.61 | 94.44 ± 9.11 | 5 (41.7%) | 1.18 ± 1.17 | 34.37 ± 17.56 | 2.31 ± 0.89 |
| *P* | **0.001** | **0.001>** | **0.001** | **0.001** | 0.129 | 0.179 | 0.303 | **0.002** | 0.387 | 0.618 |
| **Country** | | | | | | | | | | |
| USA (n = 68) | 121785.12 ± 347767.47 | 670.13 ± 1167.57 | 47.92 ± 100.85 | 59.33 ± 110.33 | 510.06 ± 534.69 | 95.09 ± 4.88 | 5 (41.7%) | 1.83 ±.15 | 36.79 ± 15.58 | 2.47 ± 0.98 |
| UK (n = 10) | 15622.20 ± 37909.14 | 39.50 ± 59.68 | 6.60 ± 14.63 | 1.20 ± 2.15 | 321.10 ± 449.52 | 88.93 ± 11.37 | 1 (8.3%) | 1.50 ± 1.58 | 44 ± 23.72 | 2.50 ± 0.84 |
| India (n = 11) | 8551.64 ± 20311.77 | 106.45 ± 180.01 | 3.54 ± 7.51 | 12.18 ± 22.28 | 690.91 ± 633.73 | 98.36 ± 2.86 | 0 (0%) | 0.72 ± 0.90 | 29.09 ± 9.65 | 2.09 ± 0.83 |
| Other (n = 11) | 40669.27 ± 42628.53 | 290. ±415.66 | 9.90 ± 14.40 | 30.45 ± 58.08 | 358.27 ± 327.42 | 96.23 ± 7.85 | 6 (50%) | 1.36 ±.92 | 31.81 ± 15.61 | 2.36 ± 1.02 |
| p | **0.008** | **0.005** | **0.016** | **0.033** | 0.082 | **0.020** | **0.001>** | **0.024** | 0.333 | 0.588 |

**Notes.**

Mann Whitney *U* test in analysis of continents, Kruskal Wallis Test test in analysis of countries.

USA, United States of America; VPI, Video Power Index; SD, Standart Deviation; GQS, Global Quality Score; JAMA, Journal of American Medical Association benchmark criteria.

Bold font: statistical significance ($p < 0.05$).

of the videos on YouTube, a popular video uploading platform that provides patients with information on choosing a TENS device before purchasing it and using the device once it is received. We found that videos created by physicians were the most common on YouTube, and videos with content related to complications were uploaded more frequently before 2018 and decreased in subsequent years. There was a statistically significant relationship between the JAMA reliability scores and the number of likes and comments. According to GQS results, 14 (14%) videos were found to be of high quality. According to JAMA results, 10 (10%) videos contained completely sufficient data, and 21 (21%) videos were highly reliable. According to reliability and quality scoring, no significant difference was found between video sources, and academic sources were found to be of longer duration.

TENS is known to be safe and well tolerated by many patient groups, with virtually no side effects. Therefore, TENS devices are often ordered by patients using the Internet and used without the help of a healthcare provider. However, TENS device manufacturers warn that patients with pacemakers should not use these devices and that patients with a history of epilepsy or pregnant women should use them with extreme caution. In addition, allergic complications and contact dermatitis are other dermatological complications that may occur at the site of electronic pad placement (*Teoli & An, 2022*). Another important issue is the differences in the ways or techniques of using TENS machines. The two most commonly used techniques are conventional and acupuncture. The conventional technique (low intensity and high frequency) is known to produce a strong tingling sensation in the painful area, whereas the acupuncture technique (high intensity and low frequency) causes muscle twitching by producing a strong painless pulsation sensation (*Johnson et al., 2019*). Acupuncture-like TENS can be used in patients who are unresponsive to traditional TENS and is recommended to be used less frequently than traditional TENS (*e.g.*, 20 min per day, three times per day) (*Mokhtari et al., 2020*). The lack of this information in patients during TENS use can lead to legitimate concerns about complications and unsuccessful treatment outcomes. Thus, it may be more useful for patients to consult a healthcare provider before doing their own research online.

Although YouTube was originally created for entertainment purposes, medical information videos has been the subject of extensive research. Because YouTube has no inherent mechanism to verify the accuracy and reliability of uploaded videos, it is possible for patients to be exposed to misinformation. It is obvious that this misinformation can negatively affect healthcare consumers. For example, it has been reported that YouTube may have the potential to change patients' beliefs about controversial issues such as vaccines (*Madathil et al., 2015*). *Lau et al. (2012)* found that videos containing negative portrayals about HPV vaccines get more views and can change healthcare consumers' attitudes towards vaccination. It can be predicted that watching these videos by individuals who are not healthcare professionals may reduce the effectiveness of the health campaign (*Lau et al., 2012*). In addition, information about drugs that have yet been approved by the Food and Drug Administration by some commercial companies are available in YouTube videos (*Sajadi & Goldman, 2011*). The use of such drugs without the control of a medical professional is worrisome, as it can cause irreversible health problems. On the other hand, YouTube is trying to take some precautions against such information pollution. As an
example, YouTube published a Medical Misinformation Policy in May 2020 to prevent public health misinformation that emerged during the COVID-19 pandemic (*Andika et al., 2021*). Accordingly, YouTube has declared that they will not allow videos that contradict the information of the World Health Organization and local health authorities (*Andika et al., 2021*). Despite YouTube policies, many videos that are not by health authorities continue to be uploaded to the website. Patients visit this website to research about their diagnoses, obtain information about healthcare products, and evaluate these products before ordering. Although users cannot place orders directly using YouTube, it remains the most widely used global advertising platform (*Jones et al., 2021*). Given its ease of access and ability to reach patients with low health literacy, it is possible that YouTube will continue to be an important resource for patients searching for health information in the future (*Ward et al., 2021*). Therefore, healthcare professionals are required to upload videos in which patients can find accurate and quality information.

In the present study, when videos were evaluated based on their sources, videos uploaded by physicians (40%) and health-related websites (25%) accounted for a significantly higher number of videos. Similarly, *Adorisio, Silveri & Torino (2021)* and *Hartnett et al. (2022)* examined YouTube videos on various medical topics and found that videos created by physicians were the most common. When the relationship between quality and reliability was evaluated according to video sources, *Oydanich et al. (2022)*. did not detect a significant difference between sources and JAMA and GQS scores, and *Hartnett et al. (2022)* found no significant difference between sources and DISCERN and GQS scores. The current study is similar to the literature in this respect. To increase the availability of online health information, physicians need to express that patients should be selective when accessing medical information from the web (*Singh, Singh & Singh, 2012*). Based on the Medical Library Association's ''User's Guide to Finding and Evaluating Health Information on the Web'', physicians can provide patients with basic guidelines for content evaluation, such as how to evaluate factual information, as well as sponsorships and disclosures (*Lee et al., 2020*). We believe that healthcare professionals should upload more reliable and high-quality videos with good scores and make them available to users. Based on the correlation analysis conducted in the present study, a weak positive correlation was found between JAMA scores and the number of views, and a moderate positive correlation between JAMA scores and modified DISCERN scores. Also a moderate to strong correlation was found between VPI scores and other user parameters (view, like, dislike, comment) except duration. Moreover, *Bahar-Ozdemir, Ozsoy-Unubol & Akyuz (2022)* reported similar results and found a correlation of DISCERN and GQS scores with video parameters. *Staunton et al. (2015)* reported a significant correlation between the JAMA scores and the number of views, whereas *Toksoz & Duran (2021)* found a positive correlation between the JAMA scores and the number of likes. Similar to our study, *Reina-Varona et al. (2022)* found a weak to moderate correlation between VPI and other user parameters except duration. The present study is similar to the literature in terms of these results. Notably, the high number of views, likes, and comments on highly reliable videos shows that viewers are looking for such videos with sufficient content and quality. Based on this, it can be concluded that YouTube should support and encourage content creators to upload more

reliable and higher-quality videos on public health-related issues, rather than focusing on commercial concerns, and make these videos more accessible in user feeds.

Based on the analysis of video content categories, it was found that 81 (81%) of the videos evaluated in the present study provided information about the procedure, followed by 79 (79%) videos providing information about indications. Similar to the present study, *Jamleh et al. (2021)* reported that 90.5% of the videos in their study contained information about the procedural steps.Our study is similar to the literature in this aspect. It has been determined that procedural explanations on how to apply the TENS device are the most mentioned in the videos compared to other topics. Another finding of the present study was that videos with content related to complications were uploaded more frequently before 2018, with the topic becoming less relevant with more recent uploads. It was also found that complications were the least mentioned topic among all videos. *Sari & Umur (2021)* evaluated YouTube videos on hallux valgus and reported that the least shared content was related to complications. Our study is similar to the literature in this aspect. However, when presenting topics related to public health through written or visual communication sources, the institutions or people presenting the content have an obligation to present impartial and balanced information on the topic.

Of the videos evaluated in the present study, 68%, 11%, and 10% were uploaded from the USA, India, and the UK, respectively. In terms of the continent, 71% of the videos were uploaded from America and only 29% were from non-American continents. Consistent with the results of the present study, *Jamleh et al. (2021)* ranked countries by number of videos uploaded, with the USA, India, and the UK forming the top three in that order. *Li, Giuliani & Ingledew (2021)* also found that most videos were uploaded from the USA. In the present study, videos from the USA had significantly higher JAMA scores higher views, likes, dislikes, comments, and animated content. Based on these results, it can be concluded that the USA is the world leader in uploading videos on various medical topics on YouTube, and new videos with high reliability scores should be uploaded by other countries to promote public health.

This study has some limitations. First, only videos that were available on YouTube within a certain period were included in the evaluation. However, it is known that YouTube's ever-changing dynamic content structure may lead to different results in a different study conducted in a different period. In addition, the exclusion criteria used during the selection of the first 100 videos also cause differences in the videos included in the study in different studies. Second, topic-related search terms can be vary and thus influence the results of the study. Therefore, we tried to maximize the quality of the results using a neutral term. Third, only videos in English were included in the study. Finally, although the two authors were blinded to each other, the subjective assessment of the videos *via* questionnaires may have impacted the results.

## CONCLUSION

The present study examining the analysis of TENS in YouTube, revealed that the videos were generally of low reliability and quality. It has been found that videos with high

reliability scores have more views and academic videos have longer durations. It was determined that there was no difference between reliability and quality scores according to video sources. It has been determined that procedural issues are mentioned the most in the videos and complications are less mentioned as the videos became more recent. There is a need for videos with high quality and reliable content to be presented to patients and users. Future studies exploring the usefulness of this types of videos in patient health may produce significant positive improvements in public health.

## ACKNOWLEDGEMENTS

The authors thank Enago for their assistance in manuscript native translation and editing.

### Funding
The authors received no funding for this work.

### Competing Interests
The authors declare there are no competing interests.

### Author Contributions
- Yüksel Erkin conceived and designed the experiments, performed the experiments, prepared figures and/or tables, authored or reviewed drafts of the article, and approved the final draft.
- Volkan Hanci conceived and designed the experiments, performed the experiments, analyzed the data, authored or reviewed drafts of the article, and approved the final draft.
- Erkan Ozduran conceived and designed the experiments, performed the experiments, prepared figures and/or tables, and approved the final draft.

### Ethics
The following information was supplied relating to ethical approvals (*i.e.*, approving body and any reference numbers):

The University of Dokuz Eylül granted Ethical approval to carry out the study within its facilities (Ethical Application Ref: 7548-GOA 2022/34-23, Date: 26.10.2022).

### Data Availability
The raw measurements are available in the Supplemental File.

### Supplemental Information
Supplemental information for this article can be found online at http://dx.doi.org/10.7717/peerj.15412#supplemental-information.

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
