# Peer review of "Evaluation of the reliability and quality of YouTube videos as a source of information for transcutaneous electrical nerve stimulation"

_PeerJ, doi:10.7717/peerj.15412_

## Round 0.1 · original submission · Major Revisions

Dear Authors

Three reviewers have analyzed your manuscript and have issued a detailed evaluation of your research. Although some reviewers rate your article as relevant research, there are important concerns regarding methodology, exposition of results, and presentation of discussion that need to be addressed in order to reevaluate your manuscript again, unfortunately at the current state of the manuscript it cannot be published. But we invite the authors to respond to the review as I believe that the issues raised by the reviewers are very pertinent and the manuscript will benefit greatly from the revision.

It is important that the results of this study are discussed in greater depth and the possible practical and clinical implications of this type of work are detailed.

Thank you for submitting your manuscript to Peerj, we look forward to receiving your manuscript with the response to the reviewers.

Reviewer 1 ·

Basic reporting

Dear author,

Your present article is interesting and its objective is useful considering the free use of TENS device without any professional prescription or supervision required. The new ways of obtaining knowledge provide independence to the users, but they also increase the risk of spread low quality or fake information. The analysis of YouTube, which is an important social network for seeking health information, is a novel project for many authors who want to contrast the quality and reliability of this information.

I congratulate you for your work. However, I have made some suggestions to improve its quality and solve some problems that I have observed. I hope these observations may be of your help:

Abstract:

Please, in the Results section (lines 40-41), you should change the weak correlation to moderate, because JAMA scores and likes showed a moderate positive correlation (r=0.566), not weak.

In the conclusion statement (lines 42-48), you should also answer the third objective regarding the comparison between user parameters and video quality and reliability.

Manuscript:

In lines 53-55, this information may have been too repeated by previous studies.

In lines 100-102, the fact that the reliability and quality of this topic are questionable should not be an assumption at the beginning of the study. It is tested by the study.

Most of the references are updated. However, some references use capital letters at the beginning of each word and other references do not. Please, keep the format of the references homogeneous.

Tables and Figures:

The structure of the article is neat and easy to read. I suggest to improve the quality of the tables and figures, because they are difficult to interpret.

In Tables 2 and 4, I do not understand very well the reasons to categorize the results of the different topics into these year ranges, because they are not proportionate.

In Tables 4 and 6, if you obtained more than two categories that were compared regarding their continuous data, the Kruskal-Wallis would have been the correct statistical test, not the Mann-Whitney U test. This problem can be observed in the years, video sources, GQS, JAMA, DISCERN, and country categories.

In Table 5, GQS, JAMA, and DISCERN scores are ordinal data. Moreover, in the previous analysis, non-parametric tests were used. Did these variables show a normal distribution? Considering the ordinal data, a Spearman correlation test would probably be more correct.

In figure 1, it is difficult to identify the categorization of the three last boxes. I would include the different scales for the quality and sufficient data categorizations. For example,

GQS

High quality (n=14)

JAMA

Completely Sufficient Data (n=10)

I would also improve the quality of this figure and Figure 2.

Experimental design

Methods:

In line 121, was the search strategy conducted independently by both authors? I recommend stating it in more detail in the manuscript. Example "two independent reviewers identified videos containing medical content using...".

In line 125, What was considered an unacceptable audiovisual quality?

In lines 128-130, this could represent a limitation due to the order of appearance of the videos, which changes every year and is different depending on the user. This could represent a limitation if valid videos were excluded. You have pointed this out in the limitations section, but you did not mention the decision of excluding those videos after the first 100 as a limitation. I think it would be better to indicate it.

In lines 157-158, it could be important to record dislikes too, and analyze them along with the other quantitative variables. There is a Chrome extension that shows you the dislikes on YouTube videos.

In lines 170-171 and Table 5 (mentioned in Table 5 suggestion), a Pearson’s correlation test was used to analyze the correlation between variables. Were these variables normally distributed to perform a Pearson correlation test? Moreover, GQS, JAMA, and DISCERN scores are ordinal variables. I suggest applicating the Spearman correlation test instead of Pearson.

Validity of the findings

Results, Discussion and Conclusion:

In lines 175-176, in the Methods section, the inclusion criterion was the first 100 eligible videos, but here you indicate that the first 100 videos with the highest number of views were included. It was not an inclusion criterion in the Methods section. You should indicate the correct inclusion criterion in both places.

In lines 194-197, it would be easier to read the information if you showed each absolute number/percentage with each respective source. For example, "41 were uploaded by physicians, 32 by health-related websites...".

In lines 198-199, 278-279, 344-345, 378-379, and Tables 2 and 4, the separation between the different year categories is not proportional. It is normal to assume that there were more videos released before 2018 than in 2019-2020 or 2020-2021, because the category 2018> contains more years.

In lines 227, 228, 327-329, correlation coefficients of 0.566 and 0.688 are not weak correlations, they are moderate. I suggest correcting them. In this study, you can see the different ranges and their interpretation based on the different r values DOI: 10.1213/ANE.0000000000002864.

In lines 233-234, US is not a continent, I think that there was some confusion here.


In lines 265-271, this information seems a bit redundant with the Introduction. It could better support the information shown in the Introduction section.

In lines 342-343 and 348-349, did you have any references that sustain these statements?

·

Basic reporting

The text is written in easy-to-understand technical English. The line of argument is coherent and the study is presented in a way that makes the reader feel interested both in the results and in the discussion of them from the introduction.

I consider the number of references to be sufficient and up to date.
Regarding the introduction, I propose that the clinical applications of TENS be separated according to the type of pain treated, since the efficacy shown in the literature is not the same in the management of nondisciplastic pain or neuropathic pain. Including different paragraphs to comment on the efficacy of this type of intervention in the different types of pain could improve the introduction.

I suggest introducing a reflection on the audience that can consult the videos (both professionals and patients), so this question could be mentioned in the introduction.

The results and discussion fit perfectly with the hypothesis put forward. The conclusions should be presented in a less direct format, leaving room for future studies to explore the usefulness of this type of video on patient health. Lines 381,382 and 383 should be reworded.

Experimental design

The design conforms to the editorial line of Peer J. The methodology used for the study is correct and both the description of the system for obtaining videos from YouTube and the scales applied comply with the intention proposed in the objectives.
The statistical analysis tools are well selected.

From an ethical point of view, the study has been approved by an ethics and research committee.

From a methodological point of view, the study is in accordance with the journal. The description of the method can be useful for novice readers and the results are easily reproducible by performing the same search on Youtube.

Validity of the findings

The results obtained are of interest to the scientific community. Considering that we are using more and more information from non-formal systems from an academic point of view, it is important to point out that some of the information is not of high quality. This aspect should be emphasized since some of these videos could generate confusion or false expectations in patients when they view them.
The information obtained is solid, but I believe that the possible negative impact of bad information should be discussed in more depth in the discussion section.
The final statement in the conclusion should be reworded, as it is not supported by data.

Additional comments

The study presented is very interesting.
I think that in view of the population that seeks information in this type of sources, it is important to analyze the content and be critical as scientists with it. The authors have made a remarkable effort in the visualization and screening of information and present results that are of enormous interest for the group of people who recommend the use of TENS to their patients.

Reviewer 3 ·

Basic reporting

The manuscript is well written and clear. The field of study is novel, original and interesting, but I have my doubts about the real impact and the clinical implications of the study. From my point of view, authors should emphasize these points. Below you can find some points and questions that I hope they could help to increase the quality of the manuscript

Experimental design

Introduction:

I think that the term “Electrotherapeutic neuromodulation techniques” (line 69) should be replaced by “electrical stimulation therapies”. However, I think that the paragraph should be start more directly with Transcutaneous Electrical Nerve Stimulation, avoiding general neuromodulation techniques and invasive techniques.

I would recommend adding information about why TENS is also used for urinary and fecal incontinence, peripheral ischemia, would healing, tissue regeneration, dementia and stroke. Furthermore, the effects of TENS on spasticity has also been explored and documented.
Methods:

There are some aspects that seems subjective or not accurate…. What are the criteria to determine that videos are irrelevant? How do you rate the “content of acceptable audiovisual quality”?

Just as a suggestion: Regarding to “video parameters”, other authors use indexes such as the Video Power Index to calculate video popularity. This more quantitative measure allows for more comparisons between different categories. See Reina-Varona et al. 2022. (https://www.ncbi.nlm.nih.gov/pmc/articles/PMC9528906/ )

Regarding “Source of videos”, from my point of view, it would be very interesting to categorize commercial videos (or videos with a commercial purpose). I think that this category is inside “health-related web sites”, but this is a wider category that could mask biased videos with commercial purposes. I think that this idea should be better addressed in the methodology and the discussion section.

Line 136. A full stop is missing at the end of the sentence

Validity of the findings

Results:
Although the first 100 eligible videos were included in this study, I think that it would be of interest to know the total number of videos found in the search. I recommend complete the information in the flow diagram, that in the current version is poor.

Line 187-88: “No significant change in animation content was observed over the years”: How do you quantify “animation content”? is just a yes/no question?

Line 204-205: “ no statistically significant correlation was found between the video sources and JAMA results” : I don`t understand how authors can perform a correlation with the “video sources” outcome. To perform a correlation is needed 2 outcomes that you can rate or quantify. However, I don´t know how you can rate or quantify the “video sources” outcome because it is a nominal outcome.
The term “follow up” is used in several times and it could be confusing. I think that authors mean “visits”, as is stated in the methods section.

Line 227-28: “a weak positive correlation was found between…”: r: 0.688 is considered a “moderate” correlation, and not a “weak” correlation.

TABLE 6: The group “other countries”, with n=11 is OK for the table, but it should be listed in the text which countries are.

Discussion:
I think that the discussion section is poor and specific results are not discussed. The main part of the discussion could be written before performing the analysis. In my opinion, the real implications of the results of this study are not exposed.
Lines294: The sentence “… a strong painless TENS sensation” is confusing. I suggest “a strong tingling sensation”

Conclusion:
I think the introductory sentence and the conclusion should be more concise and direct. I think that the results should not be repeated.

---

## Round 0.2 · Minor Revisions

Dear Authors

Your manuscript has improved significantly from the previous revision. The reviewers have shown their satisfaction with the changes made however, there are still some minor points that I feel need to be addressed before we can accept the manuscript definitively.

There are some calculations of the VPI and correlation analysis that are apparently in error and this is highlighted by one of the reviewers, please check this point.

Best regards

Reviewer 1 ·

Basic reporting

No comment.

Experimental design

No comment.

Validity of the findings

Dear authors,

I have cautiously read the manuscript and I am pleased that you have addressed most of the problems that the reviewers have mentioned. However, I have detected a serious problem with the VPI calculation that has a direct impact on the other analysis performed:

1) It seems that there is a problem with the VPI calculation. The VPI is a percentage data, so it is impossible to obtain a higher score than 100. However, in the Results section, line 240, the results regarding VPI show a 102.23 mean value, with a 71.55 standard deviation. Maybe, it has been an error during the calculation of the VPI scores. I recommend reviewing the formula used to calculate it and redoing it.

2) Moreover, the mean data (102,23) is shown with a coma instead of a period. It must be exchanged to adapt it to the English language.

3) In the results section, line 270, there is a mistake in the p-value of the dislike variable, which shows a result of p = 0474. I supposed that a period between 0 and 4 is missed (p = 0.474).

4) Finally, it is necessary to address the mistake in the VPI calculation to obtain a correct result in the correlation analysis regarding the VPI scores and the other variables. In lines 289-291, it is curious that those videos that obtained more likes have reached worse VPI (r = -0.476, p < 0.001). Taking into account that VPI calculates the ratio of likes to total likes and dislikes, it is strange that those videos with a higher number of likes have been associated with a lower VPI score, and vice versa.

I hope that this review can help you to improve your article quality.

Best regards.

Additional comments

No comment..

Reviewer 3 ·

Basic reporting

Changes and recommendations from the reviewers have been addressed successfully. The methodology section is more rigorous and the quality of the discussion has been improved. Authors have done a good job.

Experimental design

Ambiguous descriptions has been improved

Validity of the findings

The discussion section has been impreved

---

## Round 0.3 · accepted · Accept

Dear Authors

I am pleased to inform you that your article in its current state is accepted for publication. Thank you very much for submitting your manuscript to PeerJ.